# Effect of Roughness of Mussels on Cylinder Forces from a Realistic Shape Modelling

Antoine Marty [1], Franck Schoefs [2,*], Thomas Soulard [3], Christian Berhault [4], Jean-Valery Facq [1], Benoît Gaurier [1] and Gregory Germain [1]

1 Ifremer, Marine Structure Laboratory, 150 Quai Gambetta, 62200 Boulogne sur Mer, France; antoine.marty@ifremer.fr (A.M.); jvfacq@ifremer.fr (J.-V.F.); bgaurier@ifremer.fr (B.G.); ggermain@ifremer.fr (G.G.)
2 Research Institute of Civil Engineering and Mechanics, Sea and Litoral Research Institute, Ecole Centrale de Nantes, Université de Nantes, CNRS, 44322 Nantes, France
3 Laboratoire de Recherche en Hydrodynamique, Énergétique et Environnement Atmosphérique, Sea and Littoral Research Institute, Ecole Centrale de Nantes, CNRS, 44322 Nantes, France; Thomas.Soulard@ec-nantes.fr
4 Scientific Expert, Self-Employed, 83140 Six-Fours-les-Plages, France; christian.berhault22@gmail.com
* Correspondence: franck.schoefs@univ-nantes.fr

**Abstract:** After a few weeks, underwater components of offshore structures are colonized by marine species and after few years this marine growth can be significant. It has been shown that it affects the hydrodynamic loading of cylinder components such as legs and braces for jackets, risers and mooring lines for floating units. Over a decade, the development of Floating Offshore Wind Turbines highlighted specific effects due to the smaller size of their components. The effect of the roughness of hard marine growth on cylinders with smaller diameter increased and the shape should be representative of a real pattern. This paper first describes the two realistic shapes of a mature colonization by mussels and then presents the tests of these roughnesses in a hydrodynamic tank where three conditions are analyzed: current, wave and current with wave. Results are compared to the literature with a similar roughness and other shapes. The results highlight the fact that, for these realistic roughnesses, the behavior of the rough cylinders is mainly governed by the flow and not by their motions.

**Keywords:** marine growth; hydrodynamic loading; roughness; mussels; morison coefficients

## 1. Introduction

Since 2010, Floating Offshore Wind Turbines (FOWT) have been shown to be very promising for producing offshore wind energy in deeper water (>60 m) while reducing the need for spatial area nearshore where the sharing of space with other activities creates conflicts. Few prototypes and pilot farms have proven the maturity of floating concepts for wind turbines. It is now facing the reduction of cost that relies on the optimization of design, the development of new installation processes and new technologies for inspection and maintenance [1]. The need for new components in comparison with Oil and Gas floating platforms has also been shown. The size of the floater is smaller and that is the case for its underwater components: mooring lines and power cables. The order of magnitude of their diameter is 0.3–0.5 m [2,3]: these small diameters in comparison with components of Jackets offshore platforms lead to an increase of the relative roughness (i.e., ratio between the roughness and the smooth diameter) in comparison with previous tests carried out by the oil and gas industry. By increasing the role of the roughness, the effect of its shape should be reinvestigated.

Mooring lines and power cables are recognized to be the most critical components for which the feedback from the Oil and Gas industry cannot be transferred immediately [2,3]. Even if the Oil and Gas sector invested in research and development for mooring line

design, it experienced unexpected failures: according to Ma et al.'s review on failures of permanent mooring systems between 2001 and 2011 [4], the annual probability of failure was estimated to be around $3 \times 10^{-3}$ over an average sample of 300 permanent mooring systems from oil and gas industry. This assessment steered operators towards a strengthening of safety factors, which can involve solutions such as mooring lines redundancy or thicker mooring lines. However, reducing mooring system CAPEX leads to avoid redundancy, to lighten mooring lines components, to shorten their length and to use high-performance nonstandard materials such as synthetic ropes. The nascent floating offshore wind industry then faces a challenge: reducing mooring system CAPEX without increasing the risk of high consequences in case of failure. According to Fontaine et al.'s review of "past failures, pre-emptive replacements and reported degradations" (Figure 7 in [5]), over 74 analyzed failures, it became clear that fatigue is one of the main issues. Based on the same observations, the JIP led by Carbon Trust identified four major innovation needs for mooring systems ([6], p.48), among them the "Understanding of fatigue mechanisms in floating wind mooring systems". According to Braithwaite and McEvoy, offshore fish farms experienced failures due to the presence of biofouling [7]. The loading of these underwater beam components is usually modeled through the Morison quasi-static equation [8] where drag and inertia coefficients comprise as much as possible the complex hydrodynamic interactions between water and the cable. Macro-fouling, called marine growth in the following, has been shown to change drastically the value of these coefficients and thus the loading [9–11] and the structural reliability [12–14]. Three effects have been shown to drive the loading changes [10]: the change of the diameter, of the mass and of the roughness by both changing the quasi-static and the dynamic loading [15].

In 1990, Sarpkaya summarized more than 20 years of research on the effect of roughness [16]. It was shown that this changes both the bounds of hydrodynamic regimes (from laminar to turbulent) and the level of the loading. Ameryoun [17] simulated the effect of the growth of mussel's roughness through a flowchart of the load computation from the response surface model [18] and concluded it may lead to an increase of 50% of the drag force in a single year. Usually, the experimental hydrodynamic test over-simplifies the shape of the marine growth: it is usually modeled with sand or gravels. When the shape is more realistic by using a natural colonization by barnacles, anemones or seaweeds [18], the shape is not fully described as well as the surface density of specimens. It is usually summarized in a single value: relative roughness computed as the ratio between the roughness (surface to peak distance) and the smooth diameter of the component. This simplification of the real geometry explains part of the discrepancies of tests in basin reported in the standards [19,20]. Furthermore, there are only few published reports about on-site marine growth assessment from inspections and they usually register only a mean thickness and the type of species of a multi-layer marine growth [21–24]. It is thus not possible to depict a representative roughness. Recently, underwater image processing was improved [25–28] and the first quantitative data were extracted: among them, the roughness of mussels, a dominant species in Atlantic and North-Sea area [3,29]. This paper takes advantage of these data to provide a realistic reproduction of marine growth in terms of geometry and density.

With a view to simplifying further bench-marking studies, a dedicated test campaign is carried out with two homogeneous realistic shape roughnesses. The main objective of this study is to analyze how the taking into account of the real hard roughness geometry influences the loading estimation. This paper is split into four sections. Section 2 introduces the model of the two tested geometries, the experimental setup and the selected hydrodynamic conditions are chosen to be as representative as possible to those encountered by underwater mooring lines (submitted to wave and current effects). Section 3 gives the results in terms of drag and inertia Morison coefficients. Three types of conditions (current only, wave only, current plus wave) are imposed to the three studied geometries (1 smooth and 2 rough) and the results are compared with those of the literature in order to underline

the benefit to take into account the real geometry of roughness for hydrodynamic load estimation. Section 4 summarizes the paper.

## 2. Hard Marine Growth Reproduction and Experimental Setup

The objective of the tests is to represent as close as possible the hydrodynamic conditions and bio-colonization by mussels encountered by underwater components of floating offshore wind turbines. In this section, we first propose a realistic reproduction of marine growth in terms of geometry and density. The experimental setup is presented a second time.

### 2.1. Realistic Shape of Colonization by Mussels

This paper focuses on the full coverage by adult mussels encountered on Atlantic and North Sea offshore structures (species Mytilus Edulis). Recently, underwater inspections and image processing were carried out on two test sites. A day and a depth were selected, with a view to get the best conditions according to [30], and three pictures on three separated parts of a chain were taken with the aksi3D® (Figure 1c) developed during the ULTIR project [25]. In these best conditions (luminosity, turbidity, distance to the target), the accuracy reaches 0.7 cm. This chain is the main anchoring material for the buoy equipment of the SEMREV site, operated by Ecole Centrale de Nantes, where adult mussels were observed. The same type of pictures was obtained 10 km away, on the test platform UN@SEA ee (called UN-SEA-SMS previously) [3] of Université de Nantes, two years after its installation in June 2017. The organization of each specimen and the roughness were measured. Figure 1 illustrates the organization of the specimen. On Figure 1a, the red frame represents a pattern of size 20 cm × 20 cm that was shown to be representative of an elementary representative organization of the specimens on the covered surface. Figure 1b represents the top view of mussels by an elliptical shape whose major axis inclination with respect to x axis is reported in Table 1. Note that for simplifying the presentation, mussels are aligned horizontally and vertically in Figure 1b that is not the case due to the important difference between the size of vertical and horizontal axes (see position X and Y in Table 1). For simplifying future modeling and bench-marking, the major axis is approximated by 8 values: 0°, +/− 30°, +/− 45°, +/− 60° and 90°. Similar absolute value of the inclination is plotted with the same color. Table 1 gives the position of the centers and the inclination for each of the 16 specimens in the patch. It is shown that the organization is not totally random and that similar angles are observed: it comes from the fact that an optimal organization of mussels should optimize the access to food, that is, phytoplankton obtained by filtering the sea water.

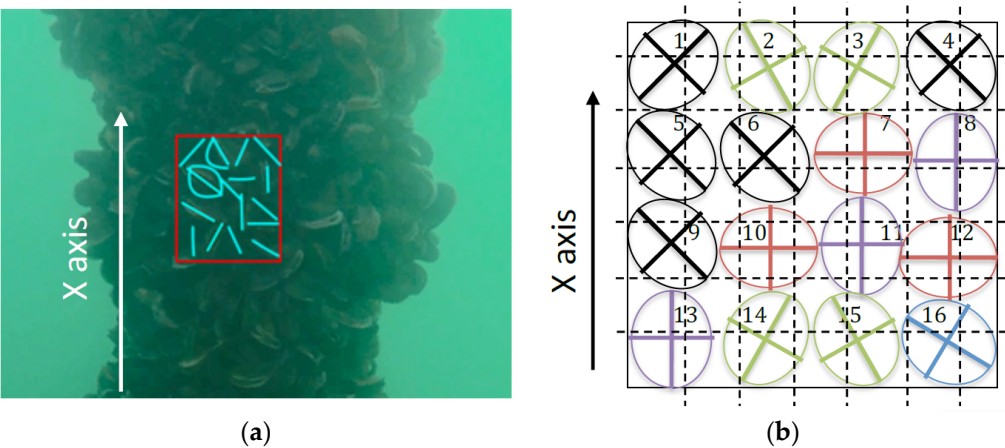

|(a)|(b)|

**Figure 1.** *Cont.*

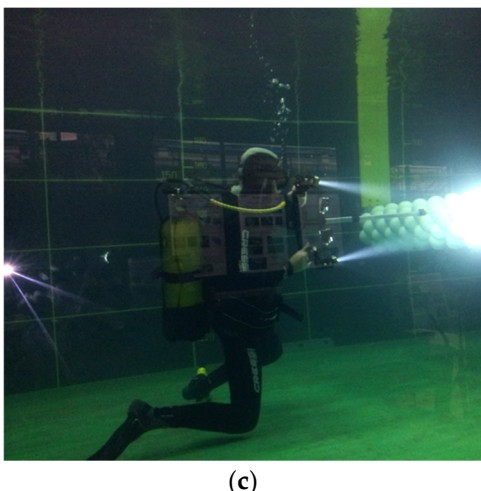

(**c**)

**Figure 1.** (**a**) Typical underwater picture with the organization of the specimen; (**b**) corresponding elliptical shape with major and minor axis: purple (0°), green (+/− 30°), black (+/− 45°), blue (+/− 60°) and red (90°); (**c**) using the aksi3D® system (tested at IFREMER).

**Table 1.** Position of the center and inclination of each specimen.

| N° Position/Angle | 1 | 2 | 3 | 4 | 5 | 6 | 7 | 8 | 9 | 10 | 11 | 12 | 13 | 14 | 15 | 16 |
|---|---|---|---|---|---|---|---|---|---|---|---|---|---|---|---|---|
| X | 8 | 25 | 41 | 58 | 8 | 24 | 43 | 59 | 8 | 26 | 42 | 58 | 8 | 25 | 41 | 58 |
| Y | 59 | 58 | 58 | 59 | 42 | 42 | 42 | 41 | 26 | 25 | 26 | 23 | 9 | 8 | 9 | 8 |
| Inclination of major axis/axis x | +45° | −30° | +30° | −45° | +45° | +45° | +90° | 0° | +45° | +90° | 0° | +90° | 0° | −30° | +30° | +60° |

Roughness was also measured and modeled according to the protocol described in [30]. Figure 2a shows the definition of a roughness $k$ and typical numbers. Note that the ratio $l/k$ varies between 1 and 1.15 for adult mussels. In this study, the value $l/k = 1.1$ is used. The relative roughness e is defined as the ratio $k/D_e$, where $k$ is the dimension of the studied roughness and $D_e$ the equivalent diameter. In the literature, several definitions of the roughness exist [10,31]. Decurey et al. [3] give a definition of $D_e$ in line with on-site measurements. In Ameryoun et al. [17], they used a stochastic modeling of marine growth and hydrodynamic parameters to define the roughness as the ratio of the apparent height of the surface roughness (mussel length from the wider section to the external extremity, $k$) on the equivalent diameter of the studied configuration. Indeed, a mussel cover may be composed of several highly compact superimposed layers. As such, layers below the external one represent a thickness of closed surfaces where no fluid dynamics is permitted, with no entrapped water volume. This closed volume corresponds therefore to the difference between the whole thickness (from the internal diameter to the extremity, $th$) and the surface roughness ($k$). Figure 3 represents the different parameters for the calculation of the equivalent diameter.

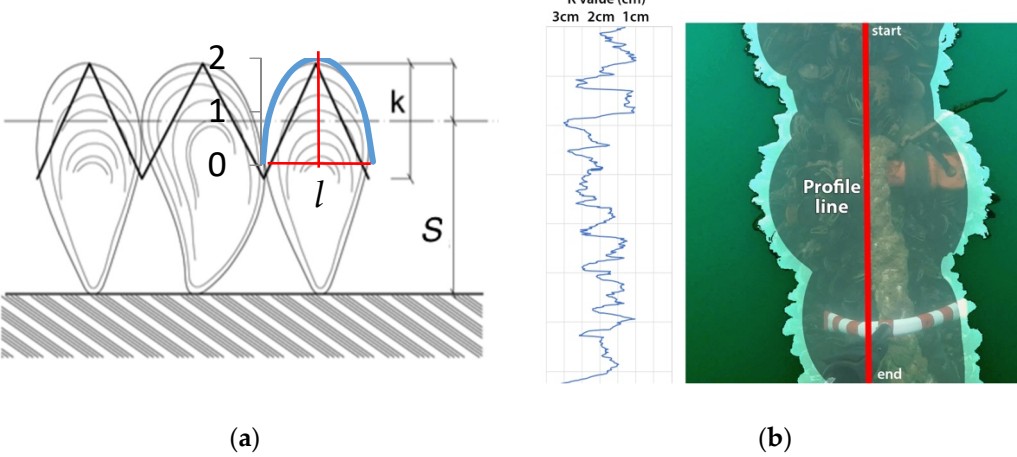

(**a**) (**b**)

**Figure 2.** (**a**) Definition of the roughness for a given size of the shell S (numbers in cm); (**b**) typical extraction of the roughness from image processing.

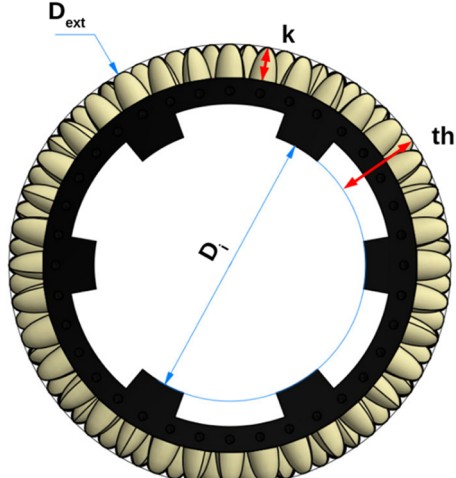

**Figure 3.** Definition of roughness parameters adapted to the experimental set-up.

Consequently, this thickness is assumed to be a diameter increase in a fluid dynamic point of view and thus the equivalent diameter is calculated as follow.

$$D_e = D_i + 2\,(th - k). \tag{1}$$

Then, the relative roughness *e* is defined only from the external layer, over a cylinder of equivalent diameter $D_e$. Applying the same principle on the external layer, the part below the wider section of the mussel is considered closed. Consequently, only the mussel height upon the wider section is considered to define the roughness k, representing the surface irregularities impacting the flow boundary layer. Several lines were inspected and Figure 4 provides the distribution of the roughness that were measured between 1.5 and 3 cm [30].

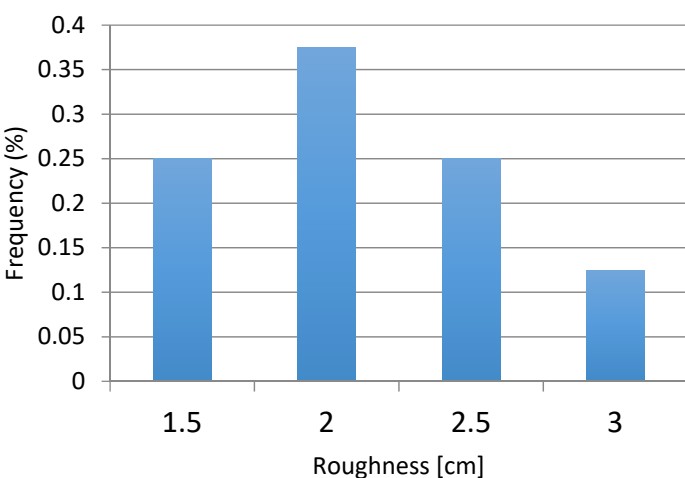

**Figure 4.** Distribution of roughness extracted from image processing.

The objectives of this paper are:

− to analyze the effect of a realistic roughness on the loading and to compare with other tests in the literature,

− to highlight whether the realistic size of mussels significantly impacts the loading.

For these reasons, two sizes are selected for the roughness: small size mussels (mussel roughness of 20 mm) and larger individual mussels (mussel roughness of 30 mm). This is to notice that when mussels cluster around the rope, they do not fill all the space and create interstices full of water. In the following, we consider that only the last level of mussels creates the surface roughness and the other levels create a close volume due to the high concentration of mussels. To this end and by means of 3D printing, two types of mussels shape and distributions have been considered. The first one is called *C1* with an outside diameter (Dext) equal to 260 mm with small size mussels (roughness of 20 mm) and the second one called *C2* with outside diameter equal to 280 mm composed of larger individual mussels (roughness of 30 mm). According to Figures 1 and 2, the design of the roughness due to mussel can be modelled as a semi-ellipsoid with the minor and major axes and its height. Both configurations have an ellipse base of 16 × 18 mm and 20 mm tall for *C1* and 24 × 27 mm and 30 mm tall for *C2*. Precise dimensions are given in the Figure 5.

For each mussel's shape, the distribution around the cylinder follows the same pattern. Mussels are arranged depending on their angle between the major axis of the ellipse and the cylinder axis according to Figure 1b in such a way as to generate a stochastic distribution network as shown on the Figure 5 Right. The eight angles pattern is repeated all along the circumference of the cylinder and then reproduced along the cylinder axis with a staggered positioning, represented with the arrows on the drawing. Note that, for printing reasons, the four specimens horizontally aligned on Figure 1b were arrayed in checkerboard; that agrees also the real organization of mussels for which there is an important difference between the size of vertical and horizontal axes. The experimental set-up is based on a smooth cylinder (called *S*) of diameter *D* = 160 mm, on which the roughness is superimposed in order to design a configuration with roughness (see Figure 6). The three cylinders' arrangement characteristics are summarized in Table 2.

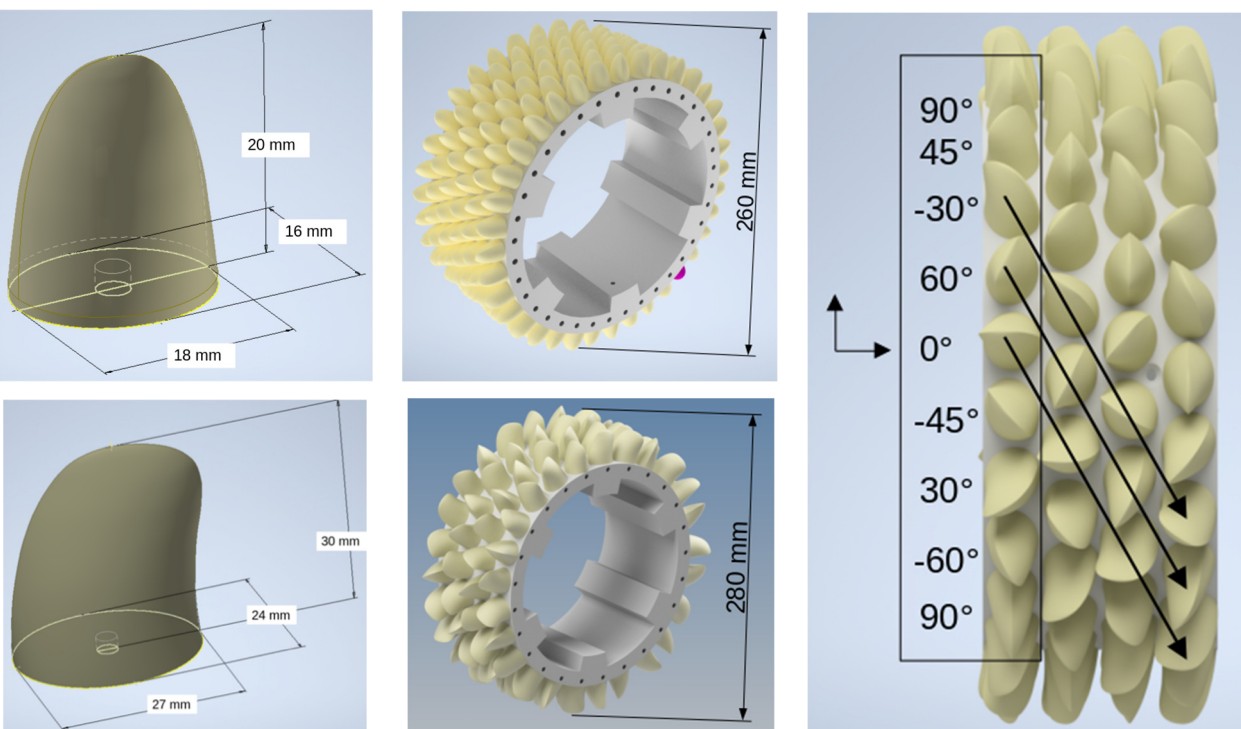

**Figure 5.** Mussels' roughness shape for C1 on top and C2 at the bottom. On the right, mussels distribution around the cylinder with the C2 shape.

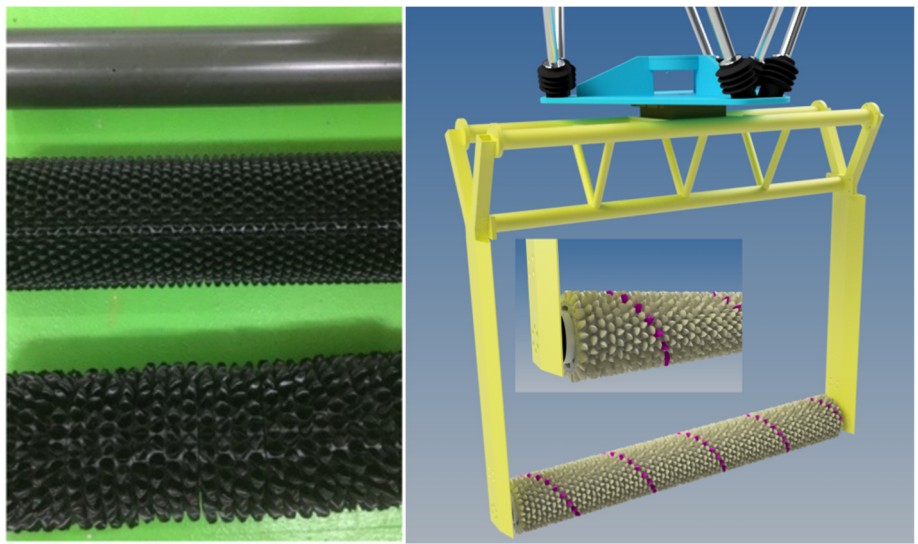

**Figure 6.** On the left, from the top to the bottom, cases *S*, *C1* and *C2*. On the right, *C2* roughness mounted on the cylinder.

**Table 2.** Synthesis of the studied roughness parameters for all configurations.

| Configurations | $D_i$ [mm] | $D_{ext}$ [mm] | k [mm] | th [mm] | $D_e$ [mm] | e = k/$D_e$ | Mass System [daN] | Areal Density for nb. Specimens/m$^2$ |
|---|---|---|---|---|---|---|---|---|
| S | 160 | 160 | 0 | 0 | 160 | 0 | 47 | - |
| C1 | 160 | 260 | 20 | 50 | 220 | 0.091 | 105 | 2969 |
| C2 | 160 | 280 | 30 | 60 | 220 | 0.136 | 110 | 1374.5 |

### 2.2. Realistic Hydrodynamic Configurations

Reynolds $R_e$ and Keulegan-Carpenter $KC$ numbers have been shown to drive the evolution of drag forces and inertia coefficient of Morison equations with the water particle velocity. Their definition in the presence of marine growth [11] is presented in Equations (2) and (3):

$$R_e = \frac{U\,D_e}{v} = \frac{A_x\omega\,D_e}{v} \qquad (2)$$

with $v$ the kinematic velocity, $U$, the flow velocity or the oscillation speed $A_x\omega$.

$$KC = 2\pi\frac{A_x}{D_e} \qquad (3)$$

The reduced speed is defined as:

$$U_r = \frac{U}{f\,D_e} \qquad (4)$$

with $f = \frac{\omega}{2\pi}$.

The objective is to cover common hydrodynamic conditions with $4.10^4 < R_e < 3.10^5$ and $4 < KC < 12$. The range of $KC$ allows to detect the strong non-linearities of drag and inertia forces with particle velocity.

According to the potential of the equipment (see Section 3), values in Table 3 are reached for each configuration.

**Table 3.** Synthesis of the normalized numbers covered for all configurations.

| Configurations | KC | $U_r$ | Re/10$^5$ |
|---|---|---|---|
| S | 3.9–15.7 | 4.1–39.1 | 0.4–2.7 |
| C1 | 2.5–11.4 | 3–56.8 | 0.55–3.8 |
| C2 | 2.5–11.4 | 3–56.8 | 0.55–3.8 |

## 3. Experimental Setup

We seek to understand the hydrodynamic behavior of a submarine cable under waves and current conditions. In this way, we performed tests using a fixed cylinder (with or without roughness) under current conditions first, then the superimposition of an oscillating cylinder under these same current conditions with the aim of reproducing the effect of the wave and current interaction. The horizontal oscillating motions of the tested cylinder, which simulate the wave part, are made using a 6-axis hexapod.

### 3.1. Ifremer Flume Tank, Assembly and Instrumentation

The tests are carried out in the wave and current circulating flume tank of Ifremer located in Boulogne-sur-Mer (France) [32]. The test section is: 18 m long × 4 m wide × 2 m high. In this work, the three instantaneous velocity components are denoted $(U; V; W)$ along the $(X; Y; Z)$ directions respectively (Figure 7). The incoming flow $(\overline{U_\infty}; \overline{V_\infty}; \overline{W_\infty})$ is assumed to be steady and constant. By means of a grid and a honeycomb (that acts

as a flow straightener) placed at the inlet of the working section, a turbulent intensity of I = 1.5% is achieved.

An overview of the global set-up is presented in Figure 7. The cylinder movements are generated using a 6-axis hexapod on which the structure and the instrumentation are fixed. As shown on Figure 7, the cylinder is horizontally freely mounted so that the cylinder is located in the middle of the test section (at one meter depth). The 2 m length cylinder is perpendicular to the direction of the upstream flow. To simulate wave conditions, the hexapode moves with an oscillating and periodic motion in parallel to the flow to represent the horizontal part of the waves' orbital velocity. The hexapode motions along the $Ox$ axis are characterized by its amplitude $A_x$ and its frequency $f$. The axis coordinate system $(x, y, z)$ is chosen so that the $Ox$ axis is in the same direction as the current. The $Oz$ axis is across the width of the basin and the $Oy$ axis is vertical and oriented upwards, see Figure 7 left.

Two 6 components load cells, with a maximal loads range of $F_{x;y;z}$ = 150 daN, fixed at each extremity of the cylinder, allow the measurement of the forces applied on the cylinder. The location of these load cells is identified by their own axis systems as shown in the Figure 7 (right). The two cylindrical load cells measure the forces applied on the cylinder only; half of the total load for each cell. The noise of the measurement is negligible. The data treatment from Morison equation requires a sinusoidal loading [33]. That explains the presence of residuals.

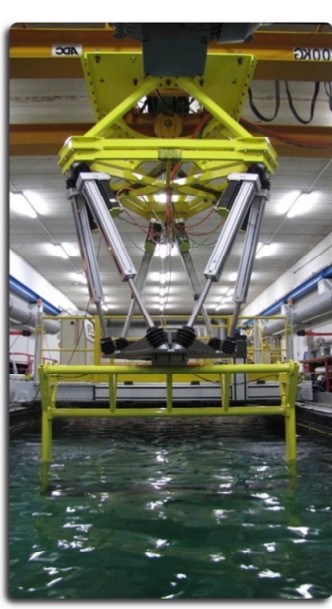
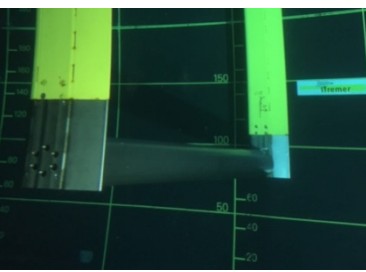
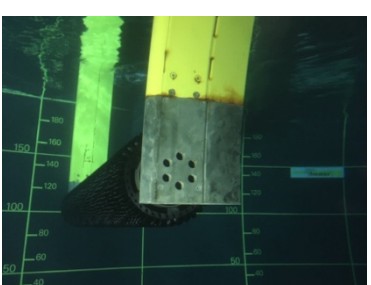
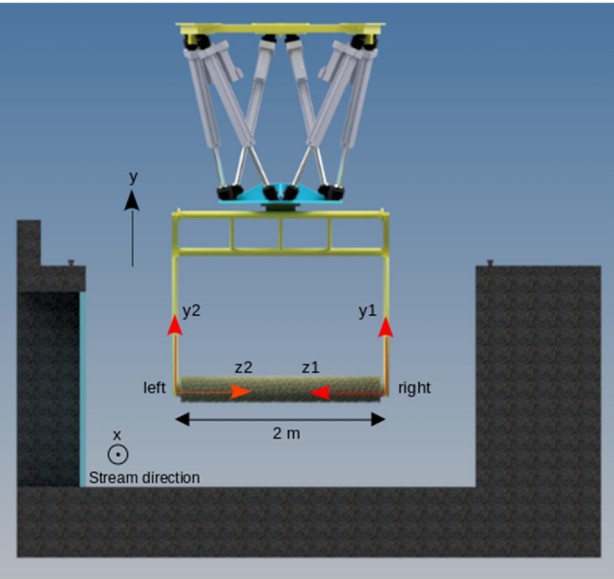

**Figure 7.** Presentation of the global set-up with the 6-axis hexapod (**left**), the smooth cylinder (**top center**) and one of the rough cylinder (**bottom center**) and axis coordinate system (x, y, z) used in tests (**left**). In black, the main system. The Ox axis is common to all systems and corresponds to the main flow direction. In red, the axes of the load cells (**right** and **left**).

### 3.2. Post-Processing of Results

The detailed procedure for computing forces and Morison coefficients is available in [34,35]. Inertia and drag coefficients are obtained. The following notations are used:

− $C_D$ for the drag coefficient in steady flow (also written $C_{DS}$ in standards)
− $C_d$ for the drag coefficient in oscillating motion (also written $C_D$ in standards)
− $C_m$ for the inertia coefficient in oscillating motion (also written $C_M$ in standards).

These three parameters are plotted as a function of the dimensionless numbers previously cited ($R_e$, $KC$ and $U_r$). All the raw data can be found on the data share platform SEANOE [34,35].

Three tests are considered and their results are commented on in the next three subsections:

$-$ Current only
$-$ Oscillating motion
$-$ Current and oscillating motion

## 4. Results and Discussion

### 4.1. Current Only Tests

Figure 8 shows the evolution of the drag forces in the function of the flow velocity. In the studied flow range, there are no drag force differences between the two roughnesses. The curves highlight the classical evolution according to a square power law for the two rough cylinders. This response is however different for the smooth cylinder with a linear evolution until the transition obtained at a flow speed of 1.25 m/s. For $U > 1.25$ m/s, the drag is quite constant around $F_D \approx 200$N.

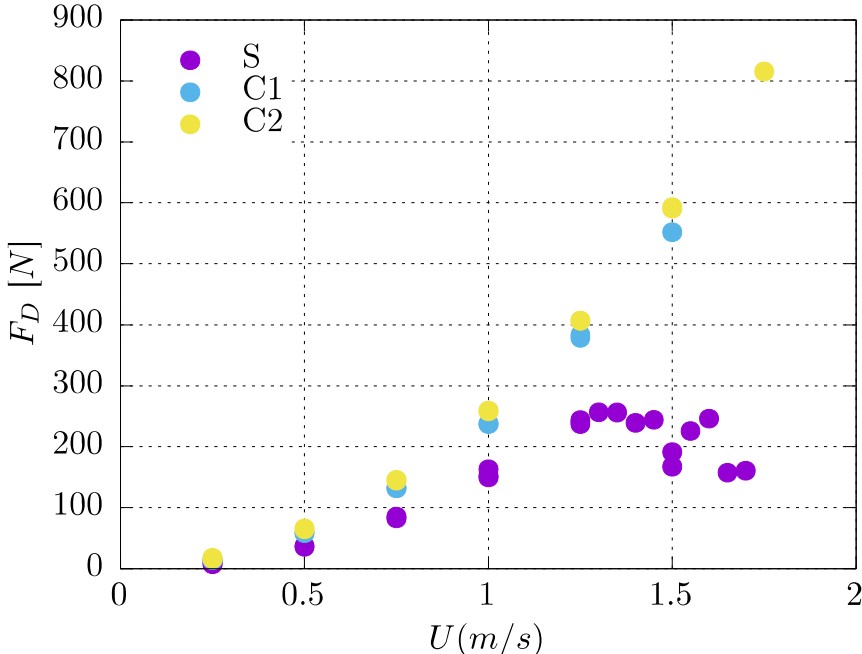

**Figure 8.** Drag force evolution for the three tested cases.

By comparing the results of smooth and shape *C2*, an increase of 58, 211 and 413% of the drag force for velocities of 1.25, 1.5 and 1.75, respectively, is observed for an increase of only 38% of diameter $D_e$. Moreover, by comparing *C1* and *C2*, the small change of the roughness increases the drag force of around 8% for velocities between 0.5 and 1.5 m/s.

Let us now focus on the variation of the overall mean drag coefficient $C_D$, denoted $C_{DS}$ in some standards, the Strouhal number $S_t = \frac{f_v D_e}{U}$ (with $f_v$ the vortex shedding frequency) and the r.m.s. values of the lift with Reynolds number (Figure 9).

For the smooth case, the overall shape of the $C_D(R_e)$ curve clearly coincides with the results presented in the literature. In the subcritical Reynolds number regime, a nearly constant value for $C_D$ of about 0.9 is found. For increasing Reynolds numbers, hence by approaching the critical flow state or lower transition that starts at $R_e \approx 2.1 \times 10^5$, this value gradually decreases. The minimum value of the drag coefficient of $C_D \approx 0.28$ at $R_e \approx 2 \times 10^5$ marks the transition from the critical Reynolds number regime to the upper transition. This phenomenon is well known [36,37] and confirms the accuracy of the experimental set-up and of the measurements.

For roughness cases (*C1* and *C2*), the transition does not occur in the flow velocity range and the relative roughness ($10^{-1}$) studied. It was observed for the smallest relative roughness between $5 \times 10^{-4}$ and $2 \times 10^{-2}$ [38]. The results show that $C_D$ increases with the size of the roughness, reaching a nearly constant value of about 1.05 for *C1* and 1.15 for

*C2*. Note that API and DNV standards gathered studies from 1971 to 1986 and recommend values of 1.11 for the relative roughness of *C1* (*e* = 0.09). However, standards do not report the results of the PhD of Theophanatos [18] (p. 96), where a discussion about similar values of e is available. In this study, cylinders fully covered by a relative roughness close to *C1* were tested (*e* = 0.085) from a single layer of mussels of size 0.27 mm with a value of $C_D$ of 1.2, close to the value obtained by pyramids and gravels. However, the areal density of the peaks was not given. For *C1* and *C2*, they are the following (Table 2):

− Areal density for *C1* = 2969 specimens/m$^2$.
− Areal density for *C2* = 1374.5 specimens/m$^2$.

It was shown that the percentage of cover (another estimate of the areal density) plays a significant role: $C_D$ varies from 1.15 to 1.2 for percentages of cover of 75% and 100%, respectively. The results of the present study suggest 1.05 instead 1.11 (standards) or 1.2 (Theophanatos), which leads to a reduction of respectively 5% and 13% of the drag force.

Moreover, standards indicate no results among 56 experiences reported in the $C_D = f(e)$ curves with a drag coefficient larger than 1.14 for the range of relative roughness $2 \times 10^{-6}$– $4.5 \times 10^{-2}$. The results of this paper show that standards could suggest a value of 1.15 for larger relative roughness up to *e* = 0.14.

It is thus crucial to report not only the relative roughness and the shape, but also the organization and areal density of the species in future studies.

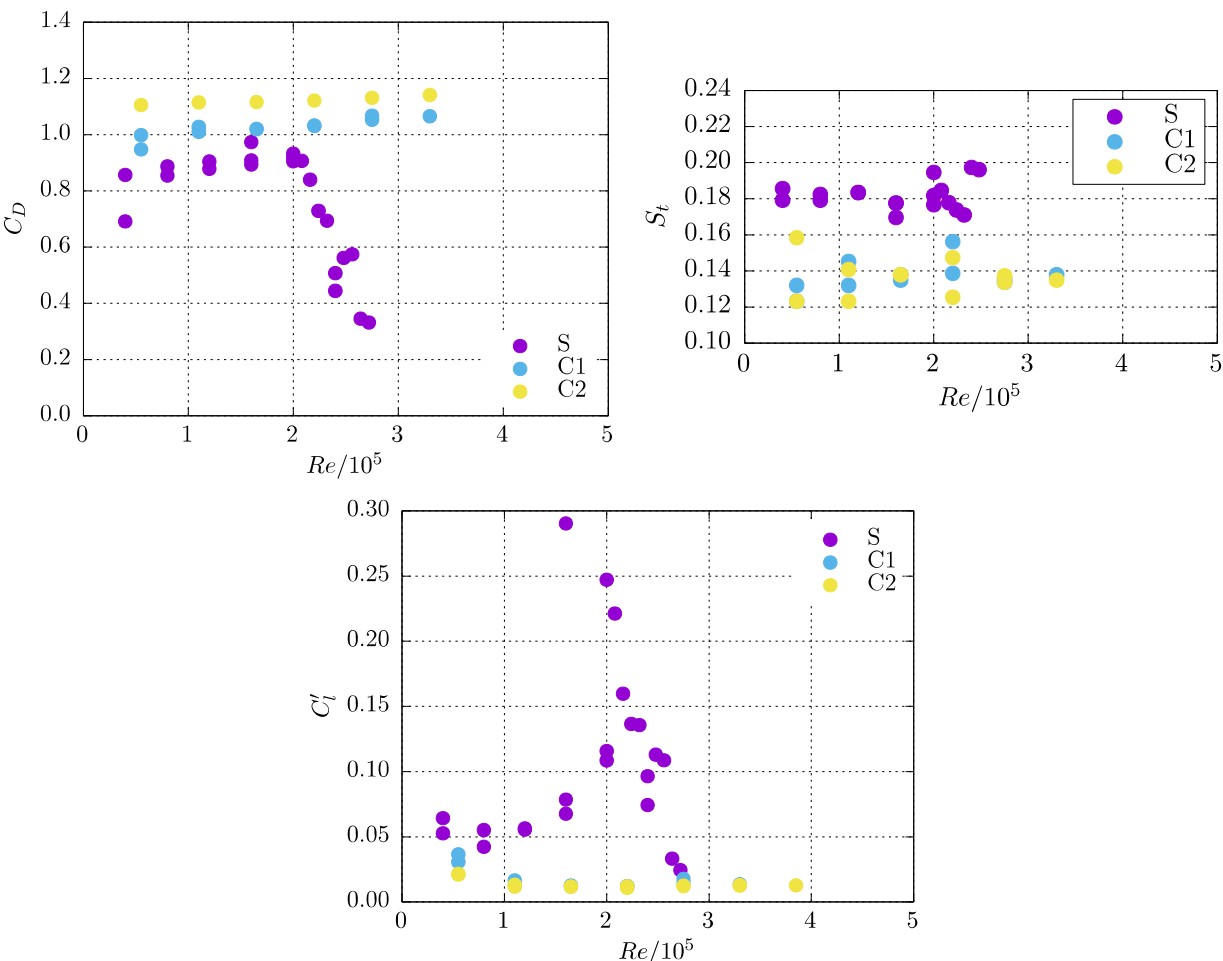

**Figure 9.** Distribution of the three main hydrodynamic parameters as function of the Reynolds number from direct force measurements for the three test cases: *S*, *C1* and *C2*.

Figure 9 presents the dependency of the Strouhal number on the Reynolds number. A constant value of $St = 0.18$ is observed in the subcritical regime for the smooth cylinder. This value is lower than the Strouhal number commonly used, which is generally equal to 0.21, see [39]. For both rough cylinders, the Strouhal number presents a nearly constant value of about 0.14.

The variation of the r.m.s values of the lift fluctuations with the Reynolds number is also shown in Figure 9. A maximum value of approximately 0.3 is obtained for $R_e \approx 2 \times 10^5$ in the subcritical state. For larger Reynolds numbers inside this flow regime, a steep decrease of the r.m.s. values is observed. For both rough cases, the fluctuations are very low with: $Cl' \ll 0.05$.

These results show that the surface roughness has an influence on the drag coefficient, the r.m.s. values of the lift fluctuations and the Strouhal number. The r.m.s. values are always lower for the rough circular cylinders. A similar trend is observed on the drag coefficient, where a difference of about 10% between cases is observed in this range ($R_e < 2 \times 10^5$). The vortices are shed into the wake with different frequencies. The Fourier transform of the lift forces shows (Figure 10) that the amplitude peaks of the vortex shedding frequencies are much higher for the smooth configuration with values of 25 N for $2 \leq R_e/10^5 \leq 2.5$ when it reaches only 2 N for the two rough configurations.

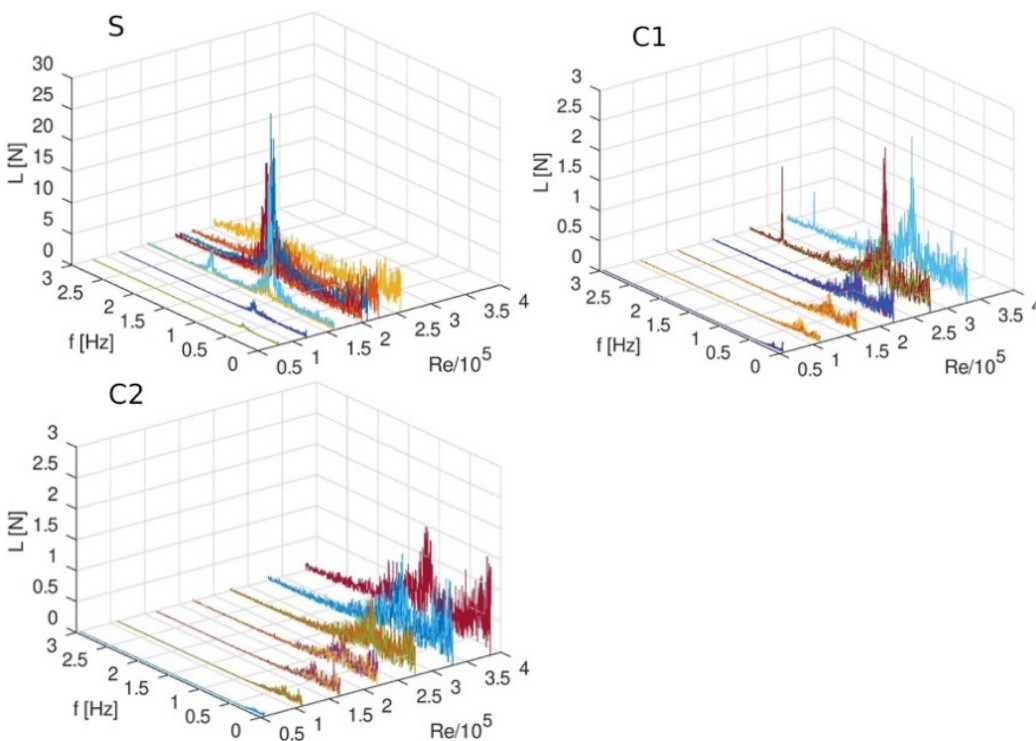

**Figure 10.** Lift forces Fourier transform as function of the Reynolds number for the three test cases: *S*, *C1* and *C2*.

The peak of the spectrum for *C1* is closer to that of *S* and is easily identified when it is more spread for *C2*. This can be explained by a two times larger areal density of mussels for *C1* in comparison with *C2* (Table 2). Configuration *C1* behaves dynamically as a smooth cylinder when turbulences appear with *C2*.

### 4.2. Oscillating Motions

For the oscillating motions test cases, the current velocity is equal to zero. Figure 11 presents the evolution of the oscillating drag coefficient $C_d$ (left), denoted by $C_D$ in some standards, and the inertia coefficient $C_m$, denoted by $C_M$ in some standards, as a function

of the Keulegan-Carpenter number *KC*. Several points are plotted per *KC* because several tests have been carried out at the same motion amplitude *Ax* but with different frequencies.

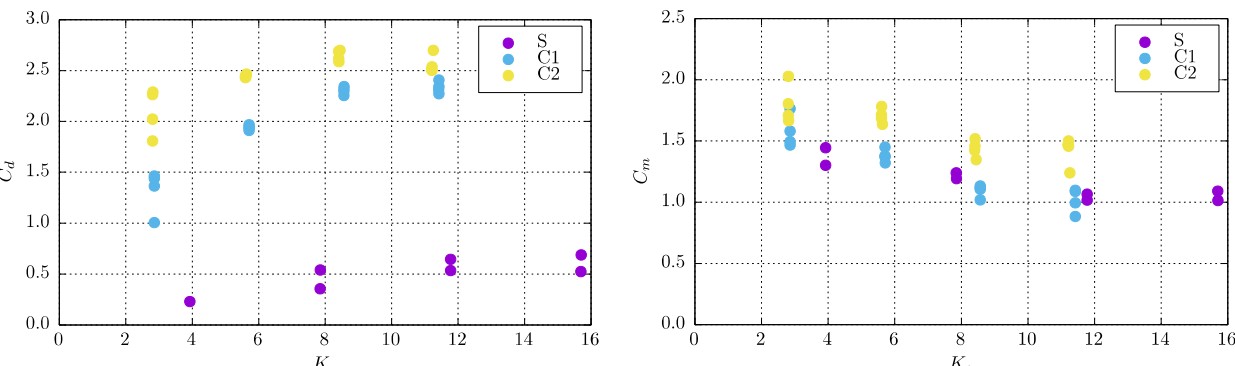

**Figure 11.** Evolution of $C_d$ (**left**) and $C_m$ (**right**) vs. *KC*.

Figure 11 (right) shows the usual trend of decreasing evolution of $C_m$ with *KC* for *KC* < 15 [16,40,41]. The results show a constant difference between the inertia coefficient of 0.3 between the two rough cylinders with a higher value for the higher relative roughness (*C2*). The $C_m$ of the smooth cylinder is slightly lower than the rough cases. To our knowledge, only the studies of Nath [42] reported values of $C_m$ for *e* = 0.1 with artificial roughness represented by cones. The values reported are significantly higher: 2.8 and 2.5 for *KC* = 5 and 12, respectively, where our test results give 1.4 and 1 for *C1*. However, again, the areal density of peaks is not reported in the paper. A realistic shape for mussels appears to drastically change the inertia coefficient.

Concerning oscillating drag coefficients between smooth and rough cases, results show a significant difference. The calculated coefficients are more than three times higher for cases *C1* and *C2* compared to the smooth case, with $C_d \approx 2.5$ for *KC* > 6 for the rough cases and $C_d \approx 0.5$ for *KC* ≤ 16 for the smooth cylinder. The behavior of the rough cylinders is mainly governed by the flow and not by their motions, contrary to the smooth cylinder for which its behavior is mainly governed by its motions. Again, only the studies of Nath were carried out with a relative roughness close to ours (*C1*): they are compared with other studies in [18] (Figure 9.9). Again, the values reported are significantly higher: 3.2 and 2.7 for *KC* = 5 and 12, respectively, with large scatters where our tests give 1.7 and 2.3 for *C1*. The effect of roughness is shown to be significant, especially for low *KC* (≈3) where $C_d \approx 1.3$ for *C1* and 2.1 for *C2*, leading to a 62% increase of drag forces.

### 4.3. Current and Oscillating Motions

This section presents results concerning current and oscillating motions tested cases. The coefficients introduced in the previous sections are calculated: the mean drag coefficient $C_D$, the oscillating drag coefficient $C_d$ and the inertia coefficient $C_m$. These coefficients are at first presented configuration by configuration as a function of $U_r$ in Figure 12. It is first observed that mean and oscillating drag are very close for both roughness cases. The mean drag coefficients are two times higher for the rough cases than for the smooth one. These results confirm that the behavior of the rough cylinders is mainly governed by the flow and not by their motions, contrary to the smooth cylinder for which its behavior is mainly governed by its motions. Inertia coefficients for the rough cases present less dispersion than for the smooth cylinder and show a value for *C2* 25% higher than for *C1* for $U_r$ < 10.

Figure 13 presents each coefficient for the three studied configurations. In order to compare the behavior of each configuration, the current velocity is fixed at 1 m/s. These coefficients are represented as a function of the reduced speed for all the motion amplitudes in order to study the amplitude and the frequency parameters effects at the same time.

The results show several and opposite behaviors of the coefficients. First of all, the inertia coefficient $C_m$ tends to be similar for each configuration. The higher the frequency

(small $U_r$), the lower the coefficient. Moreover, the motion amplitude has no impact on the evolution of the inertia coefficient. Regarding drag coefficients $C_d$ and $C_D$, their behaviors are totally the opposite. The value of $C_d$ increases with the reduced velocity $U_r$. Moreover, for a fixed frequency (or $U_r$ fixed) the amplitude parameter has a high impact and the value of the coefficient increases when the amplitude $A_m$ decreases. The exact opposite phenomenon occurs concerning the mean drag coefficient $C_D$, with the value of the coefficient decreasing when the amplitude Am increases.

Finally, as for the previous case, there is an important difference concerning oscillating drag coefficients and mean drag coefficients between smooth and rough cylinders. The calculated coefficients are much higher for cases *C1* and *C2* compared to the smooth case for which there is no dependency on the motion amplitude and frequency. A strong dependency on the amplitude of the drag coefficients at a fixed frequency for the rough cases is here clearly highlighted.

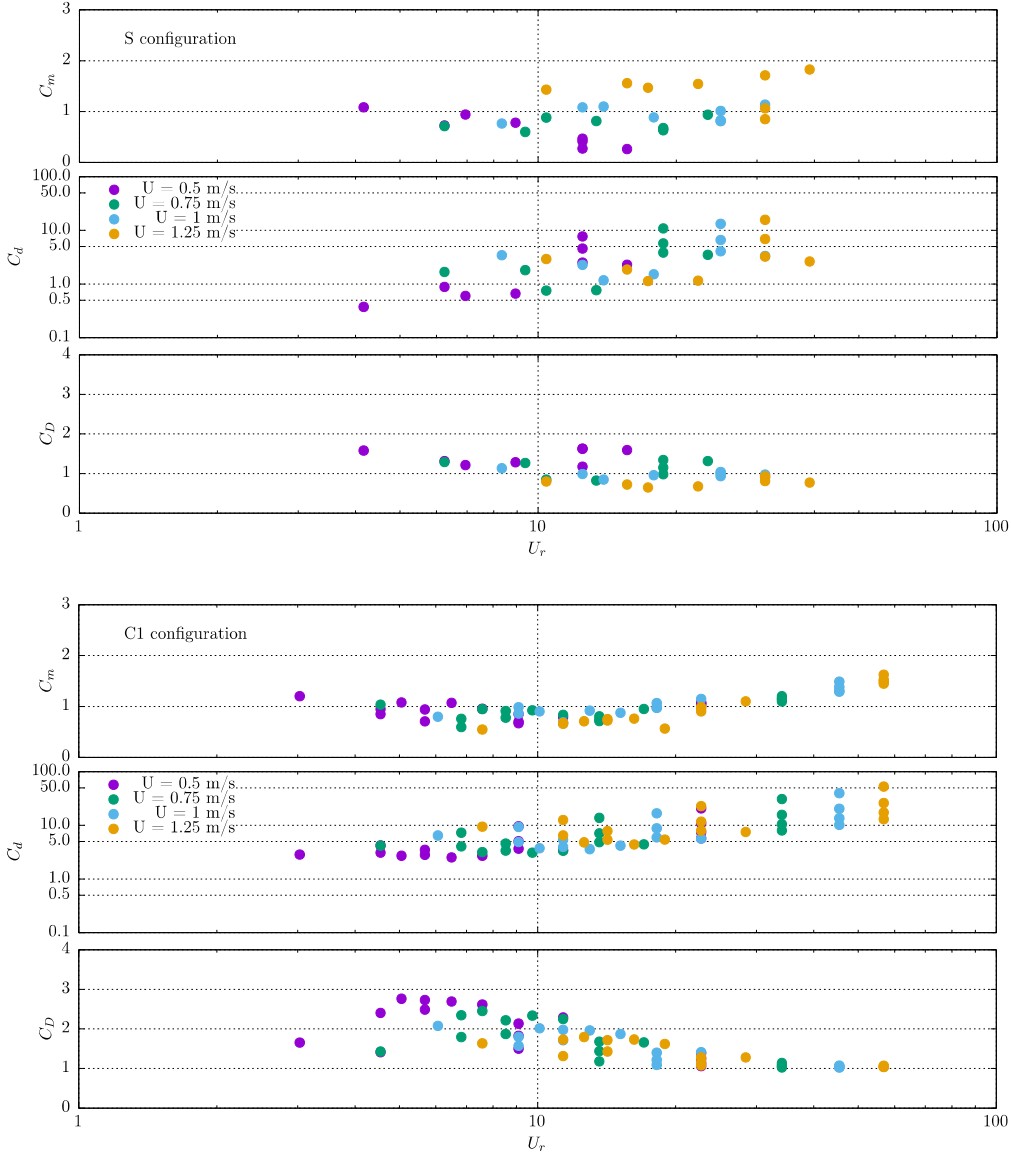

**Figure 12.** *Cont.*

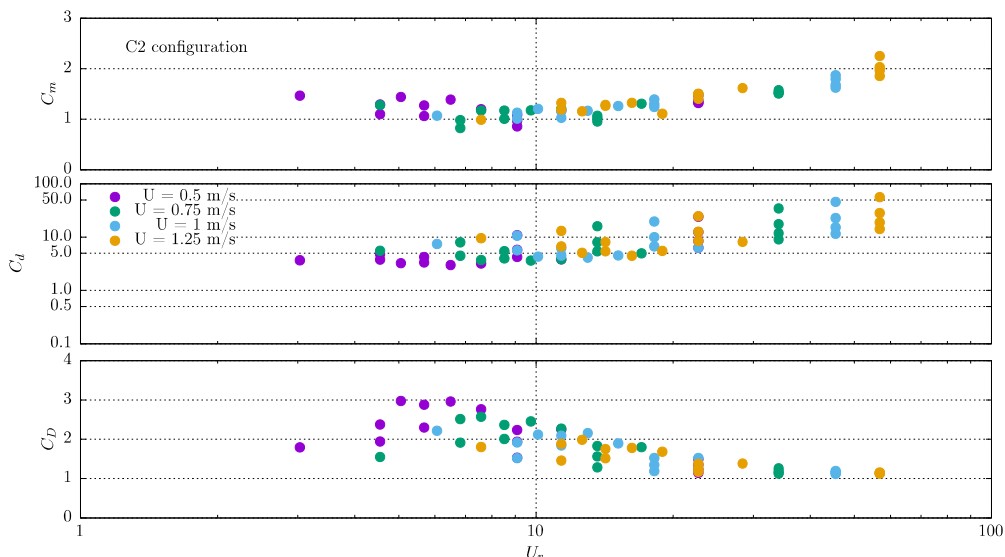

**Figure 12.** Evolution of $C_m$, $C_d$ and $C_D$ vs. $KC$ for the S (**top**), *C1* (**middle**) and *C2* (**bottom**) cases.

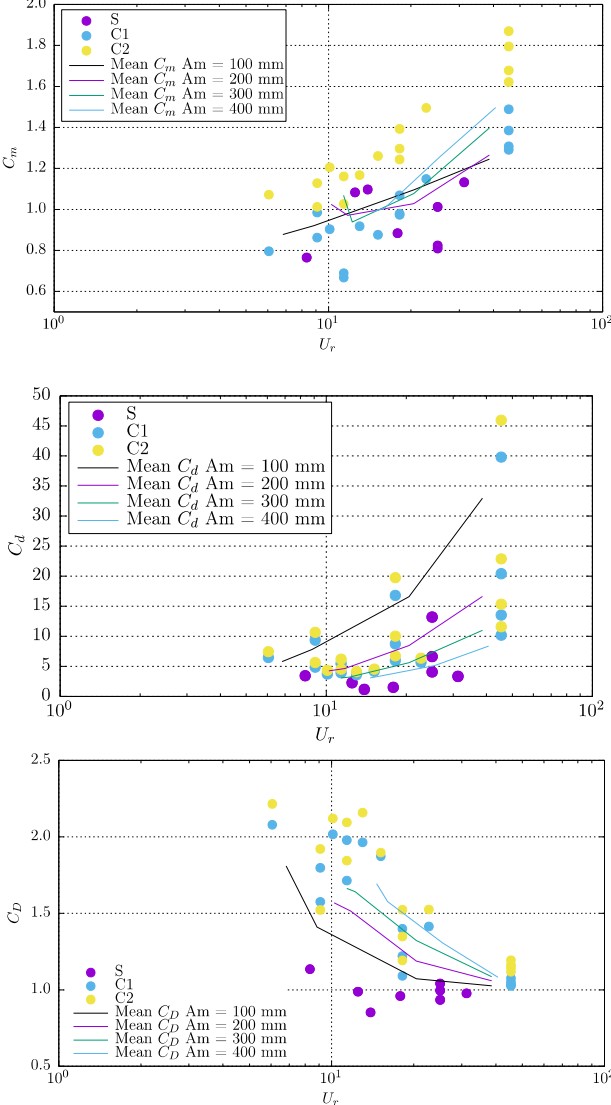

**Figure 13.** Evolution of $C_m$, $C_d$ and $C_D$ vs. $U_r$ for the *S*, *C1* and *C2* cases.

## 5. Discussion and Conclusions

This study shows the impact of two realistic hard marine growth roughness *C1* and *C2* on the drag and inertia coefficients compared to a smooth case. The shape and organization of species are deduced from on-site observations of a full colonization by adult mussels that induce high relative roughness (0.09 for *C1* and 0.14 for *C2*) for mooring lines and power cables of Floating Offshore Wind Turbines. These results show that the surface roughness has an influence on the drag coefficient, the r.m.s. values of the lift fluctuations and the Strouhal number. The results from this experimental campaign highlight significant differences concerning the forces on the two rough cylinders. For instance, the effect of roughness is shown to be significant especially for low *KC* ($\approx$3) of oscillating motion, where $C_d \approx 1.3$ for *C1* and 2.1 for *C2*, leading to a 62% increase of drag forces. The results show a constant difference between the inertia coefficient of 0.3 between the two rough cylinders with a higher value for the higher relative roughness (*C2*). The vortices are shed into the wake with different frequencies and different amplitudes, the amplitude peaks of the vortex shedding frequencies are much higher for the smooth configuration than the rough configuration with a difference of about 90%. The r.m.s. values are always lower for the rough circular cylinders. A difference of about 10% between cases on the drag coefficient is observed for $R_e < 2 \times 10^5$. For the oscillating cases, the inertia coefficients for the rough cases present less dispersion than for the smooth cylinder. For $U_r < 10$, the mean drag coefficients are two times higher for the rough cases than for the smooth one. In this case, a strong dependency on the amplitude of the drag coefficients at fixed frequency for the rough cases has been highlighted, while they are stable in static. This shows that the commonly used approach of $C_d = \psi(KC)$. $C_D(R_e)$ is not legitimate. Moreover, while Morison's linearization for static drag force is justified, it means that it is not for oscillating cases, the *KC* defined only with the amplitude is not representative of the flow variety, this number should also depend on the frequency. These results highlight the fact that the behavior of the rough cylinders is mainly governed by the flow and not by their motions, contrary to the smooth cylinder for which its behavior is mainly governed by its motions. Moreover, the results have been compared with similar studies carried out for high relative roughness (0.1); significant differences have been observed due to the fact that the shape of the rough cylinders is not well described in these studies: key information about the organization and the areal density of peaks are usually not given.

For now, assumptions are strong: the marine growth is considered to be of a homogeneous circumferential and length volume. Due to internal and inter-species competition, it has been observed that mussels may be arranged in a bulbous manner. This phenomenon has not been studied here but the roughness variations must be studied to be compared to homogenous cover, which is considered in the engineering design phase. Pure current, regular forced oscillations and superimposed loadings must be tested in order to conclude on the validity of extracted coefficients but also on the standard hydrodynamic loading's formulation commonly used.

**Author Contributions:** Conceptualization, F.S.; methodology, A.M., G.G., C.B., T.S., F.S.; tests conception, A.M., J.-V.F., B.G., G.G., C.B., T.S., F.S.; tests realization and treatment, A.M., J.-V.F., B.G., G.G.; test analysis, A.M., J.-V.F., B.G., G.G., C.B., T.S., F.S.; writing—original draft preparation, F.S., A.M., G.G.; writing—review and editing, A.M., G.G., C.B., T.S., F.S. All authors have read and agreed to the published version of the manuscript.

**Funding:** Authors are grateful to Christian Berhault (senior Consultant) and Françoise Dubois (naval Energies), for their help in LEHERO-MG project. This work was carried out within the project LEHERO-MG (Load Effect of HEterogeneous ROughness of Marine Growth) granted by WEAMEC, West Atlantic Marine Energy Community with the support of Région Pays de la Loire and in partnership with Naval Energies. This work benefits from funding from France Energies Marines as well as the French National Research Agency under the Investments for the Future program bearing the reference ANR-10-IEED-0006-28 and within the project OMDYN-2. The authors would like to thank Guillaume Damblans from France Energies Marines. This project was partly financially supported by the European Union (FEDER), the French government, IFREMER and the

region Hauts-de-France in the framework of the project CPER 2015–2020 MARCO. We are most grateful to Thomas Bacchetti for his participation conducting all the test campaign corresponding to about 800 cases.

**Institutional Review Board Statement:** Not applicable.

**Informed Consent Statement:** Not applicable.

**Data Availability Statement:** Data are shared in a consortium and will be published as soon as the agreement will permit it. They can be obtained under this permission by asking to gregogy.germain@ifremer.fr or franck.schoefs@univ-nantes.fr.

**Conflicts of Interest:** The authors declare no conflict of interest.

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
