# Peer review of "Effect of Roughness of Mussels on Cylinder Forces from a Realistic Shape Modelling"

_jmse, doi:10.3390/jmse9060598_

Round 1
Reviewer 1 Report
The author prepared a valuable manuscript for investigating the effect of roughness of marine biofouling on underwater cylinder structure. The experimental results are of good value. However, there are some issues that should be solved before the publication can be recommended.
- Line 33-34: The authors state that “Few prototypes and pilot farms have proven the maturity of this technology”. The technology is not very clear here.
- Line 38-39: “The order of magnitude…….diameter ratio and increasing effect of the shape.” Why the order leading to high roughness – diameter ratio? Please explain it clearly.
- There are some symbols “?” in Figure 1(b) and Figure2 (a). What do they mean?
- Could you raise the resolutions of Figure 2, Figure 3, Figure5, Figure 6, Figure8, and Figure 11? All the Tables should be shown in a three-line table.
- It is recommended to change the background of Figure 4 to white.
- Figure 7 is needless here.
- Line 242: It remains unclear to the reviewer how the hydrodynamic coefficients are obtained.
- Is there noise in data collection? Did the authors remove the noise in post-processing?
- Because the status of the references [29] and [32] are “under review”, it is recommended to remove them.
Author Response
Comments and Suggestions for Authors
The author prepared a valuable manuscript for investigating the effect of roughness of marine biofouling on underwater cylinder structure. The experimental results are of good value. However, there are some issues that should be solved before the publication can be recommended.
The authors acknowledge the reviewer for the positive review and scientific interest of the paper. Corrections are in red in the paper.
- Line 33-34: The authors state that “Few prototypes and pilot farms have proven the maturity of this technology”. The technology is not very clear here.
The reviewer is right. “maturity of this technology” has been replaced by “maturity of floating concepts for wind turbine”.
- Line 38-39: “The order of magnitude…….diameter ratio and increasing effect of the shape.” Why the order leading to high roughness – diameter ratio? Please explain it clearly.
The reviewer is right and the last part of the sentence in more a concluding remark. It has been clarified by replacing the sentence by: “these small diameters in comparison with components of Jackets offshore platforms lead to an increase the relative roughness (i.e. ratio between the roughness and the smooth diameter) in comparison with previous tests carried out by the oil and gas industry. By increasing the role of the roughness, the effect of its shape should be reinvestigated.”.
- There are some symbols “?” in Figure 1(b) and Figure2 (a). What do they mean?
It is probably due to the type of editor used for opening the file because it is not visible in the word version edited by JMSE office and downloaded.
- Could you raise the resolutions of Figure 2, Figure 3, Figure5, Figure 6, Figure8, and Figure 11? All the Tables should be shown in a three-line table.
We hope that the resolution was sufficient for the review and the resolution of all the figures has been raised and the tables have been changed.
- It is recommended to change the background of Figure 4 to white.
The background is in white. The change of color is probably due to the software used for the edition of the paper.
- Figure 7 is needless here.
We agree that is not a key figure for the paper because the axes were described on Figure 8 (now figure 7) and it has been deleted.
- Line 242: It remains unclear to the reviewer how the hydrodynamic coefficients are obtained.
A citation has been added. This paper explains in detail the computation of the coefficients : Gaurier, B., Germain, G., Facq, J., Baudet, L., Birades, M., and Schoefs, F. (2014). Marine growth effects on the hydrodynamical behaviour of circular structures. Proceedings of the 14th Journées de l’Hydrodynamique, Val de Reuil, France.
- Is there noise in data collection? Did the authors remove the noise in post-processing?
Noise is negligible. The data treatment from Morison equation requires a sinusoidal loading. That explains the presence of residuals (Sarpkaya, 1981). That is explained now on section 3.1.
- Because the status of the references [29] and [32] are “under review”, it is recommended to remove them.
Ref 29 was accepted if revised and that has been corrected.
Ref 32 has been replaced by: Gaurier, B., Germain, G., Facq, J., Baudet, L., Birades, M., and Schoefs, F. (2014). Marine growth effects on the hydrodynamical behaviour of circular structures. Proceedings of the 14th Journées de l’Hydrodynamique, Val de Reuil, France.
Reviewer 2 Report
Authors of the paper deal with the modelling of mussel colonies formed on underwater surfaces of offshore structures. It is an interesting case study, especially as the formation of mussel colonies can significantly impact the performance of such structures. The manuscript is well written and the research procedures seems sound. There are, however, some points that need to be clarified:
- Some details on how the image of mussels were obtained underwater should be added to section 2.1.
- Lines 122-124 should be deleted.
- Some details about the accuracy of the numerical representation of mussels shape should be added to the manuscript in section 2.1.
- Better quality of Figs. 7 and, especially, 11 would be appreciated.
- Some discussion of the cases of structure failures due to forming of mussels would be welcome addition to the Introduction section.
Author Response
Authors of the paper deal with the modelling of mussel colonies formed on underwater surfaces of offshore structures. It is an interesting case study, especially as the formation of mussel colonies can significantly impact the performance of such structures. The manuscript is well written and the research procedures seems sound. There are, however, some points that need to be clarified:
The authors acknowledge the reviewer for the positive review and scientific interest of the paper. Corrections are in red in the paper.
- Some details on how the image of mussels were obtained underwater should be added to section 2.1.
Figure 1c was added and the corresponding text for illustrating the tool.
- Lines 122-124 should be deleted.
These lines were deleted.
- Some details about the accuracy of the numerical representation of mussels shape should be added to the manuscript in section 2.1.
The accuracy of the device has been added: it has been quantified at 0.7 cm.
- Better quality of Figs. 7 and, especially, 11 would be appreciated.
We hope that the resolution was sufficient for the review and the resolution of all the figures has been raised.
- Some discussion of the cases of structure failures due to forming of mussels would be welcome addition to the Introduction section..
To the knowledge of authors, there are some events (fish farm) for which the causes of failures were an accumulation of mussels but they have not been published. Only one paper underlines this effect and is cited now: Braithwaite, R,A and McEvoy, L,A. Marine biofouling on fish farms and its remediation. Adv Mar Biol. 47:215-52. doi: 10.1016/S0065-2881(04)47003-5, 2005.
Round 2
Reviewer 1 Report
Thank you for your response. The manuscript has been improved. I have no further questions.